# Predictors of Long-Term Mortality in Patients with Stable Angina Pectoris and Coronary Slow Flow

**DOI:** 10.3390/medicina59040763

**Published:** 2023-04-14

**Authors:** Sukru Aksoy, Dilaver Öz, Melih Öz, Mehmet Agirbasli

**Affiliations:** 1Department of Cardiology, Dr. Siyami Ersek Training and Research Hospital, University of Health Sciences, Istanbul 34668, Turkey; 2Department of Cardiology, Faculty of Medicine, Medeniyet University, Istanbul 34722, Turkey

**Keywords:** coronary slow flow, long-term mortality, outcomes, stable angina pectoris

## Abstract

*Background and Objectives*: Coronary slow flow (CSF) is an angiographic phenomenon characterized by the slow progression of an injected contrast agent during diagnostic coronary angiography in the absence of significant stenosis. Although CSF is a common angiographic finding, the long-term outcomes and mortality rates are still unknown. This study aimed to investigate the underlying causes of mortality over a 10-year period in patients diagnosed with stable angina pectoris (SAP) and CSF. *Materials and Methods*: This study included patients with SAP who underwent coronary angiography from 1 January 2012 to 31 December 2012. All patients displayed CSF despite having angiographically normal coronary arteries. Hypertension (HT), diabetes mellitus (DM), hyperlipidaemia, medication compliance, comorbidities, and laboratory data were recorded at the time of angiography. Thrombolysis in myocardial infarction (TIMI) frame count (TFC) was calculated for each patient. The cardiovascular (CV) and non-CV causes of long-term mortality were assessed. *Results:* A total of 137 patients with CSF (93 males; mean age: 52.2 ± 9.36 years) were included in this study. Twenty-one patients (15.3%) died within 10 years of follow-up. Nine (7.2%) and 12 (9.4%) patients died of non-CV and CV causes, respectively. Total mortality in patients with CSF was associated with age, HT, discontinuation of medications, and high-density lipoprotein cholesterol (HDL-C) levels. The mean TFC was associated with CV mortality. *Conclusion:* Patients with CSF exhibited a notable increase in cardiovascular-related and overall mortality rates after 10 years of follow-up. HT, discontinuation of medications, HDL-C levels, and mean TFC were associated with mortality in patients with CSF.

## 1. Introduction

Coronary slow flow (CSF) was first described by Tambe et al. [1]. CSF is characterized by the slow progression of an injected contrast agent during coronary angiography in the absence of significant stenosis. It is observed in 1–7% of coronary angiograms and is more common among young people and smokers [2,3]. Particularly, CSF is typically observed in patients who undergo angiography for acute coronary syndrome, usually unstable angina. Nearly two-thirds of patients with CSF presented with an acute coronary syndrome [2]; however, a small percentage (8%) of patients may present with acute myocardial infarction [4]. CSF increases the risk of recurrent chest pain, recurrent hospitalizations, recurrent cardiac catheterizations, life-threatening arrhythmias (e.g., Torsades de Pointes), and sudden cardiac death [5,6]. CSF can also cause myocardial ischemia and recurrent acute coronary syndrome (ACS) [2,4].

Despite being a well-known angiographic finding, the exact pathogenic mechanisms of CSF are not fully understood; however, endothelial dysfunction, microvascular abnormalities, and atherosclerosis have been suggested to play a role in the pathogenesis of CSF [7]. Long-term outcomes and mortality in patients with CSF remain to be elucidated. Thus, the aim of this study was to determine the causes of long-term (10-year) mortality in CSF patients.

## 2. Materials and Methods

### 2.1. Study Population

The study enrolled patients who had stable angina pectoris (SAP) and CSF. All coronary angiographies performed between 1 January 2012 and 31 December 2012 were reviewed for the presence of CSF. All patients who underwent coronary angiography for SAP at the Siyami Ersek Thoracic and Cardiovascular Surgery Centre from 1 January 2012 through 31 December 2012 who fulfilled the criteria for a diagnosis of CSF were included. CSF was diagnosed according to the criteria described below [3]. All patients had normal coronary arteries on angiography. Past medical histories, physical examination findings, medications, electrocardiograms (ECG), and echocardiograms were obtained. Patients with ACS were excluded.

A total of 137 patients (93 males) with angiographically normal coronary arteries and CSF were included in this study. Patients with visible plaques on angiography were excluded. Other exclusion criteria were a history of valvular heart disease, congenital heart disease, coronary and peripheral artery disease, atrial fibrillation, aortic aneurysm, revascularization, coronary artery ectasia, heart failure, ACS, pulmonary embolism, chronic obstructive pulmonary disease, acute or chronic infections, cancer, autoimmune or inflammatory diseases, thyroid disease, the administration of medication with anti-inflammatory properties, hepatic or renal dysfunction, left ventricular dysfunction, and left ventricular hypertrophy on echocardiography.

Electronic medical records were used to obtain data on symptoms, physical examination findings, laboratory test results, recurrent hospitalizations, and medications, as well as ECG and echocardiography reports. Ten-year follow-up status was determined by inviting patients to the hospital by phone. Follow-up medical histories were obtained, and physical examinations and ECGs were performed. Compliance with physician follow-up and medications was determined. Information regarding deceased patients was obtained from their relatives. Comorbidities and causes of death were extracted from the hospital’s electronic medical records, the national health system database, and death certificates.

Demographic data such as age and sex, as well as medical history data pertaining to patients’ histories of hypertension (HT), diabetes mellitus (DM), hyperlipidaemia, and smoking; genetic predisposition; compliance with medications; and comorbidities were obtained at the time of angiography. Laboratory data were recorded on pre-prepared study forms. Glucose (mg/dL), triglyceride (mg/dL), total cholesterol (mg/dL), low-density lipoprotein cholesterol (mg/dL), high-density lipoprotein cholesterol (HDL-C) (mg/dL), and haemoglobin (g) levels were obtained at the time of angiography. Patients were considered hypertensive if they had a previous diagnosis of HT or used any antihypertensive medications. A prior history of DM or treatment with antidiabetic medication was used to confirm the diagnosis of DM. The study was approved by the ethics committee of the University of Health Sciences, Dr.Siyami Ersek Thoracic, and Cardiovascular Surgery Training and Research Hospital (Number: E-28001928-604.01.01-207661794-2023), and informed consent was obtained from all patients or the family member.

### 2.2. Coronary Angiography

The digital angiographic system used for performing coronary angiography was the AXIOM Sensis (Siemens AG, Munich, Germany). The decision to perform coronary angiography was made based on the results of non-invasive stress tests or a high clinical suspicion for coronary artery disease (CAD). All patients underwent selective coronary angiography using the standard Judkins technique with JL4 and JR4 6 French catheters. The access site was the femoral artery in all cases (Ten years ago, all cases in our hospital were performed through femoral access for economic reasons). All patients received Iopromide (Ultravist 370; Schering AG, Berlin, Germany) as the preferred contrast agent. The average injection volume was 5–9 mL of opaque material. The coronary arteries were visualized at a rate of 15 frames/second—from both right and left oblique positions—using cranial and caudal angles.

### 2.3. CSF and Thrombolysis in Myocardial Infarction (TIMI) Frame Count (TFC)

We defined CSF based on the criteria described by Beltram, including (a) absence of obstructive epicardial CAD, (b) TFC > 27 frames, and (c) delayed distal vessel contrast opacification of epicardial coronary arteries [3].

Two independent and experienced interventional cardiologists, who were blinded to the study, reviewed all angiograms, and the TFC was calculated for each patient. TFC was first defined by Gibson and refers to the number of cine frames required for the contrast material to reach the distal landmarks. These distal landmarks included the distal bifurcation of the LAD (known as the “moustache”), the distal bifurcation of the longest lateral left ventricular wall artery branch for the circumflex artery (Cx) (Figure 1), and the first posterolateral artery branch of the right coronary artery (RCA) (Figure 2) [8].

CSF was defined as TFC > 2 standard deviations from the published normal range. Given that the LAD is longer than the other coronary arteries, the TFC of the LAD was divided by 1.7 and a corrected TFC (CTFC) was calculated [8]. The CTFC values were utilized in further analyses. The mean TFC was calculated by taking the average of the LAD, Cx, and RCA TFCs.

Right anterior oblique projections with caudal angulation (RAO caudal view) were used for the LAD and Cx. A left anterior oblique projection with cranial angulation (LAO cranial view) was used for the RCA. The normal TFC values were 36.28 ± 2.6 frames for the LAD, 22.28 ± 4.1 frames for the Cx, and 20.48 ± 3 frames for the RCA [8]. These values were described when cineangiography was performed at an acquisition rate of 30 frames/second. Given that the acquisition rate in our study was 15 frames/second, we multiplied the TCF values by 2 to adjust our values to the 30 frames/second values [8]. 

### 2.4. Statistical Methods

The study population’s general characteristics were obtained by conducting descriptive analyses. The Shapiro–Wilk test was used to determine the distribution of the variables; a normal distribution was observed for all variables. Therefore, independent two-sample *t*-tests were conducted to compare continuous variables with two groups. Continuous variables were compared using one-way analysis of variance (ANOVA) with Scheffe’s post-hoc test. A multiple logistic regression model was used to identify covariates associated with cardiovascular (CV) and non-CV death in patients with CSF. The mean ± standard deviation was used to present continuous variables. The Chi-squared test was utilized to compare categorical variables. Counts and percentages were used to present categorical variables. *p*-values < 0.05 were considered significant. Analyses were performed with SPSS software, version 23.0 (IBM Corp.; Armonk, NY, USA).

## 3. Results

A total of 137 patients with CSF (93 males, 68%; mean age: 52.2 ± 9.36 years) were included in this study. Table 1 displays the demographic data and clinical characteristics of the study population. Twenty-one patients (15.3%) experienced long-term mortality. Nine patients (7.2%) died from non-CV causes whereas 12 patients (9.4%) died from CV causes. The causes of CV death included myocardial infarction, *n* = 3 (14.3%); heart failure, *n* = 2 (9.5%); aortic dissection, *n* = 3 (14.3%); cerebrovascular accident, *n* = 2 (9.5%); sudden cardiac death, *n* = 1 (4.8%); and bleeding, *n* = 1 (4.8%). The non-CV causes included cancer, *n* = 6 (28.6%) and COVID-19, *n* = 3 (14.3%). Table 1 presents a comparison of the demographic and clinical characteristics among survivors, patients who experienced CV death and those that died of non-CV causes. Most patients (59.8%) had a history of cigarette smoking and 34.6% of patients discontinued their medications. All patients were prescribed medications, including acetylsalicylic acid, angiotensin-converting enzyme inhibitors (ACEI) or angiotensin receptor blockers, beta-blockers, and statins. Clopidogrel and trimetazidine were prescribed to 47 and 56 patients, respectively.

An examination of recurrent hospital admissions revealed that 31 patients (22.6%) were readmitted to the hospital due to ACS. Additionally, 10 patients (7.3%) were readmitted to the hospital due to SAP. Among patients with recurrent hospitalization, coronary angiography, percutaneous coronary stenting with angiography, and coronary artery bypass graft (CABG) surgery was performed in 17 patients (12.4%), 8 patients (5.8%), and 2 patients (1.5%), respectively.

When we compared deceased and living patients, significant differences were found with respect to age (*p* < 0.001), HT status (*p* = 0.02), DM status (*p* < 0.001), discontinuation of medications (*p* < 0.001), hyperlipidaemia (*p* = 0.03), HDL-C levels (*p* = 0.005), triglyceride levels (*p* = 0.035), and smoking duration (*p* < 0.001).

Significant differences in LAD (*p* < 0.001), Cx (*p* = 0.002), RCA, and mean TFC in patients with CV vs. non-CV causes of death were observed (Table 2).

Multiple logistic regression analyses were conducted to determine the independent predictors of long-term cardiovascular (CV) mortality and total mortality among patients with CSF. Total mortality was associated with age (odds ratio [OR]: 1.220; 95% confidence interval [CI]: 1.085–1.371; *p* = 0.001), HT (OR: 18.253; 95% CI: 1.515–219.8; *p* = 0.02), discontinuation of medications (OR: 0.013; 95% CI: 0.002–0.103; *p* < 0.001), and HDL-C levels (OR: 0.853; 95% CI: 0.747–0.975; *p* = 0.019).Mortality increased 1.2-fold with every one-year increase in patient age, 18.2-fold in patients with a history of HT, 75.2-fold in patients who were non-compliant with medications, and 1.17-fold with every 1 mg/dL decrease in HDL-C levels (Table 3).

In the multiple regression model of patients with CSF, mean TFC was also associated with CV mortality (OR: 3.318; 95% CI: 1.101–10.000) (Figure 3). Specifically, the risk of CV-related mortality increased 3.3-fold with every 1-frame increase in mean TFC (Table 4).

## 4. Discussion

Our study revealed that patients with CSF have a high prevalence of comorbidities, CV risk factors, and CV-related mortality after 10 years of follow-up. To the best of our knowledge, this study is the first to examine long-term mortality in patients with CSF. In patients with CSF, 7% experienced CV-related mortality, whilethe all-cause mortality rate was 15.3%. Recurrent myocardial infarction, ACS, hospitalizations, and coronary interventions were frequently observed. The determinants of total mortality were age, HT, DM, discontinuation of medications, and low HDL-C levels. Notably, medication non-compliance was the most important factor determining CV-related mortality in patients with CSF. Other significant determinants of CV mortality included low HDL-C levels and age. 

Patients with CSF frequently presented with ischemia and recurrent chest pain. The pathogenesis of CSF remains to be elucidated; however, several studies suggest that atherosclerosis, inflammation, and endothelial dysfunction are underlying mechanisms of CSF [7]. 

The current study addresses the limitations of current luminology practices. CSF, atherosclerosis, and CAD exhibit common CV risk factors and clinical presentations. Therefore, increased long-term mortality in patients with CSF is expected [2]. CSF is a common finding on diagnostic coronary angiography (1–7% of all coronary angiographies) [2]. In patients who underwent CAG for chest pain, CSF is a simple, valuable, and additional biomarker that comes at a low cost [9]. In our study, eight patients required implantation of coronary stents for severe CAD and two patients required a CABG procedure during the long-term follow-up period. CSF is reportedly a risk factor for sudden cardiac death, arrhythmia, myocardial ischemia, ST elevation, and myocardial infarction [7,10,11]. CSF is also more common in males, cigarette smokers, and patients with increased CV risk [3]; however, the long-term mortality rate of patients with CSF is still unknown. 

While patients with CSFchest pain and normal arteries are generally considered to have a favourable prognosis, some studies have suggested that these patients may be vulnerable to ACS [3,4,12]. Specifically, in some study cohorts, approximately two-thirds of patients with CSF presented with ACS [2]. In our study, we observed that 27 patients (19.7%) were readmitted to the hospital due to ACS. Our observations indicate that CSF is associated with recurrent coronary events and poor survival, particularly in patients who are non-compliant with medications. Atak et al. demonstrated that patients with CSF have an increased rate of QT dispersion, which is associated with the risk of ventricular arrhythmias and CV mortality [13]. In addition, sudden cardiac death due to ventricular arrhythmias has been reported in patients with CSF [6,13]. These findings are consistent with our results, as one patient died from sudden cardiac death in our study.

The definite mechanism of CSF is still uncertain; however, endothelial dysfunction and increased microvascular resistance may play a role. Previous studies have reported significant differences in fractional flow reserve values and pressure between the proximal and distal coronary arteries, indicating higher resistance in the micro-circulation [14]. In addition, Fineschiet al. used invasive hemodynamic measurements to show that resting microvascular resistance is elevated in patients with CSF [15]. Beltrame et al. found evidence that CSF is linked to a sustained increase in resting coronary microvascular tone [16]. Several studies have also demonstrated that patients with CSF exhibit impaired endothelium-dependent flow-mediated dilatation, suggesting that coronary vascular endothelial dysfunction could be an important mechanism underlying CSF pathogenesis [14,17]. The presence of microvascular disease in patients with CSF has been demonstrated through myocardial biopsy studies [18]. In these studies, endothelial thickening and lumen stenosis were observed. Therefore, CSF may represent an early, subclinical form of atherosclerosis [7]. Indeed, two studies utilizing advanced imaging techniques, such as intravascular ultrasound, have demonstrated that CSF may indicate the presence of diffuse, non-obstructive atherosclerotic disease in the coronary arteries [19,20]. The findings from the present study support this hypothesis; over time, indistinct atherosclerotic plaques can become obstructive atherosclerotic plaques and lead to vascular occlusion, particularly in patients who are non-compliant with their medications. 

In this study, we observed that patients who discontinued their medications hadhigher mortality rates. Specifically, the most important factor determining CV-related causes of death in patients with CSF was discontinuation of medications. These findings demonstrate that the treatment and follow-up of patients with CSFshould be equivalent to that of patients with CAD. In this study, nearly 40% of patients haddiscontinuedtheir medications during follow-up, likely contributing to the etiology of recurrent coronary events and poor survival. Previous studies have shown that non-compliance with cardioprotective medications (e.g., ACEI, statins, and beta-blockers) among outpatients with CAD was associated with elevated risk of mortality from all causes [21,22]. Compliance with medical treatment plays a crucial role in achieving favorable clinical outcomes in patients with CAD, such as a decreased re-admission rate and reduced CV-related morbidity and mortality [23].

Another determinant of CV and all-cause mortality in this study was low HDL-C levels. In the literature, an association between HDL-C levels and longevity has been proposed [24]. Likewise, other studies have revealed that high HDL-C levels prevent many age-related diseases, whereas reduced levels of HDL-C are associated with an increased risk of coronary heart disease [25]. Indeed, epidemiologic studies have revealed a negative linear correlation between HDL-C levels and mortality [26]. Moreover, in a meta-analysis, Gordon et al. demonstrated that for every 1 mg/dl (0.026 mmol/L) increase in plasma HDL-C levels, the risks of CHD and CV-related mortality were reduced by 2–3% and 3.7–4.7%, respectively [26]. The Framingham Study also revealed an inverse correlation between HDL-C levels and both CV disease and total mortality [27]. The anti-inflammatory, antioxidant, anti-apoptotic, anti-thrombotic, and vasodilatory properties of HDL-C may explain its atheroprotective characteristics [28,29]. In accordance with the literature, HDL-C levels were correlated with CV-related mortality and total mortality in the present study. Consistent with our results, Hawkins et al. demonstrated that patients diagnosed with CSF had significantly lower HDL-C levels [30]. In light of these findings, it is reasonable to conclude that patients with CSF should be encouraged to make lifestyle modifications, such as smoking cessation, participation in regular physical activity, and consumption of a healthy diet—in addition to receiving medical treatment.

Several clinical observational studies demonstrate that hyperglycaemia is associated with poor outcomes in patients with obstructive CAD [31,32]. On the other hand, patients with angiographically normal coronary arteries with slow flow represent a population with a potentially different etiology and pathophysiology. Recent studies define MINOCA (Myocardial Infarction with Non-Obstructive Coronary Arteries) as a separate clinical entity [33,34,35]. MINOCA and Ischemia with No Obstructive Coronary Artery Disease (INOCA) are commonly observed findings after coronary angiography [36]. 

The association between poor clinical outcomes and high blood glucose appears to be different in obstructive versus non-obstructive CAD. In patients who present with myocardial infarction and obstructive coronary arteries (MIOCA), hyperglycaemia is associated with poor outcomes [37]; however, the prognostic impact of hyperglycaemia in INOCA/MINOCA remains controversial. 

In our study, in patients with normal coronary arteries, the history of DM did not affect the presence or the severity of CSF. Further studies are required to explore the differences in the pathophysiology of MINOCA, INOCA, and MIOCA. Both INOCA and CSF are heterogeneous clinical entities. Multiple risk factors such as inflammation and thrombosis are likely to interplay to determine the poor outcomes.Similar to our findings, MINOCA patients display poor outcomes with a substantial risk of recurrent major adverse cardiac events (MACE) during follow-up [38]. In patients with MINOCA, there is evidence of pro-thrombotic activity and coronary microvascular dysfunction [39].

Our study adds to the knowledge about CSF outcomes in the literature in patients with CSF—in that these patients experienced poor survival. The limitations of our study include a small sample size and differences in the prevalence of risk factors such as age, HT, and DM among groups. Moreover, since we could not reach all recorded patients, we could not create an age- and sex-matched control group. CSF is a very complex phenotype that extends beyond known CAD risk factors such as thrombosis and inflammation. Novel indices and biomarkers are required to improve our understanding of CSF. Furthermore, larger randomized controlled studies are required to confirm our observations.

## 5. Conclusions

Patients with CSF experienced high CV-related mortality and total mortality after 10 years of follow-up. Age, HT, DM, discontinuation of medications, and low HDL-C levels were risk factors for mortality in CSF patients. Our findings support the notion that coronary angiography findings extend beyond luminology. Further studies are required to confirm CSF as a therapeutic target in cardiovascular disease.

## Figures and Tables

**Figure 1 medicina-59-00763-f001:**
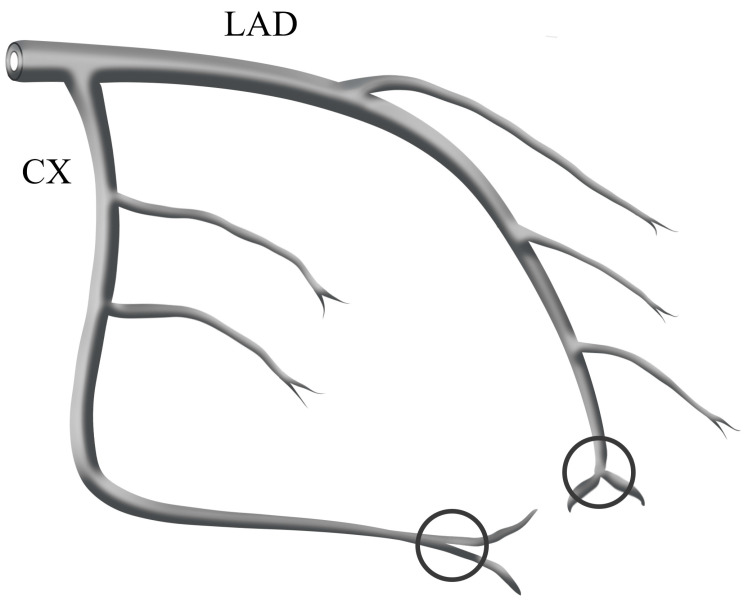
The schematic diagram shows distal landmark points for LAD and Cx. These include the distal bifurcation of the LAD, also known as the ‘moustache’, as well as the distal branch of the lateral left ventricular wall artery farthest from the coronary ostium for the Cx.

**Figure 2 medicina-59-00763-f002:**
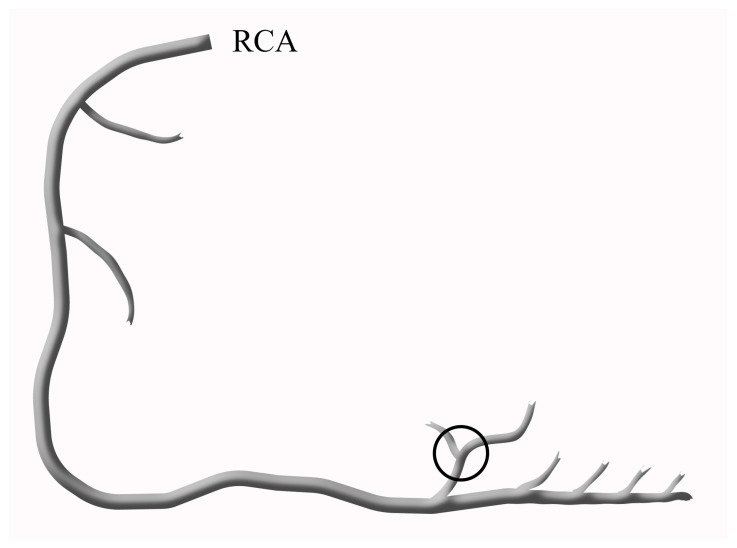
The schematic diagram of the RCA shows the distal landmark, which is defined as the first posterolateral artery branch of the RCA.

**Figure 3 medicina-59-00763-f003:**
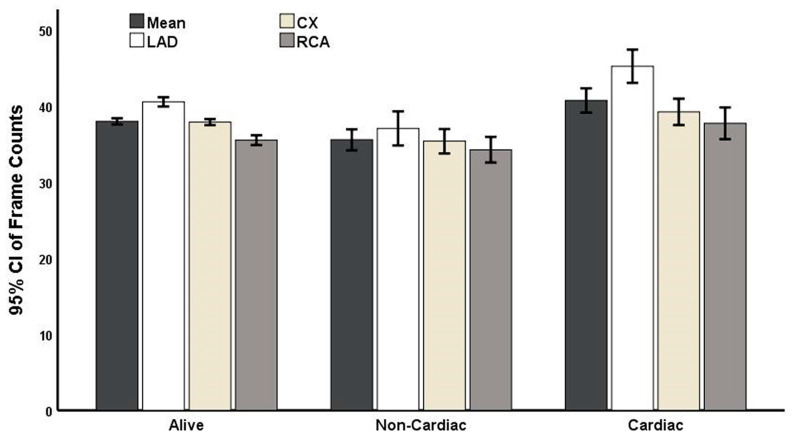
Causes of deaths as well as comparisons of the thrombolysis in myocardial infarction (TIMI) frame counts of surviving patients and those who experienced cardiovascular or non-cardiovascular death after 10 years of follow-up.

**Table 1 medicina-59-00763-t001:** Study population and comparisons of demographic and clinical characteristics among patients who are alive, those who experienced cardiovascular deaths, and those who experienced non-cardiovascular deaths.

		Study Population (*n* = 137)	Alive (*n* = 116)	Non-Cardiovascular Mortality (*n* = 9)	Cardiovascular Mortality (*n* = 12)	*p*
Gender (Male)		93 (67.9)	77 (66.4)	6 (66.7)	10 (83.3)	0.487
Hypertension		92 (67.2)	73 (62.9) ^a^	7 (77.8) ^a,b^	12 (100) ^b^	0.026
Diabetes mellitus		43 (31.4)	29 (25) ^a^	6 (66.7) ^b^	8 (66.7) ^a,b^	0.001
Family History of atherosclerosis		6 (4.4)	5 (4.3)	0 (0)	1 (8.3)	0.650
Non-compliance with medical treatment		46 (33.6)	29 (25) ^a^	7 (77.8) ^b^	10 (83.3) ^a,b^	<0.001
Hyperlipidaemia		51 (37.2)	48 (41.4)	1 (11.1)	2 (16.7)	0.059
Smoking Status	Non-smoker	55 (40.1)	44 (37.9)	6 (66.7)	5 (41.7)	0.136
	Smoker	58 (42.3)	48 (41.4)	3 (33.3)	7 (58.3)	
	Ex-smoker	24 (17.5)	24 (20.7)	0 (0)	0 (0)	
Recurrent Hospitalizations	None	100 (73)	90 (77.6) ^a,b^	7 (77.8) ^a^	3 (25) ^b^	<0.001
	Acute Coronary Syndrome	31 (22.6)	20 (17.1)	2 (22.2)	9 (75)	
	Stable Angina	10 (7.3)	10 (8.6)	0 (0)	0 (0)	
Intervention	None	110 (80.3)	91 (78.4)	9 (100)	10 (83.3)	0.324
	Angiography	17 (12.4)	17 (14.7)	0 (0)	0 (0)	
	Angiography + stenting	8 (5.8)	6 (5.2)	0 (0)	2 (16.7)	
	Bypass surgery	2 (1.5)	2 (1.7)	0 (0)	0 (0)	
Age (years)		52.2 ± 9.36	50.86 ± 9.04 ^a^	58.67 ± 8.87 ^b^	60.33 ± 6.83 ^b^	<0.001
Duration of smoking (pack-years)		20.59 ± 6.54 (*n* = 82)	19.31 ± 5.77 ^a^ (*n* = 72)	31 ± 3.61 ^b^ (*n* = 3)	29.29 ± 4.03 ^b^ (*n* = 7)	<0.001
Total Cholesterol (mg/dL)		204 ± 38.79	202.98 ± 36.21	220.31 ± 38.73	201.6 ± 59.75	0.427
HDL-C (mg/dL)		42.08 ± 9.44	43.04 ± 9.58 ^a^	39.33 ± 7.6 ^a,b^	34.83 ± 5.27 ^b^	0.010
Triglyceride (mg/dL)		173 ± 65.89	170.05 ± 70.27	181.56 ± 32.66	195.08 ± 24.16	0.424
LDL-C (mg/dL)		126.59 ± 35.07	125.07 ± 32.12	144.67 ± 38.74	127.75 ± 55.07	0.271
Hemoglobin (g/dL)		14.22 ± 1.33	14.24 ± 1.32	14.12 ± 0.75	14.16 ± 1.77	0.955
Glucose (mg/dL)		103.07 ± 21.96	101.96 ± 19.82	99.11 ± 10.02	116.75 ± 39.37	0.072
Survival time (years)		9.41 ± 1.74	10.00 ± 0.00	7.11 ± 2.32	5.42 ± 2.84	<0.001
LAD TFC *		40.67 ± 3.66	40.49 ± 3.31	37 ± 2.92	45.17 ± 3.43	<0.001
CX TFC		37.78 ± 2.33	37.83 ± 2.18 ^a^	35.33 ± 2.06	39.17 ± 2.72 ^a^	0.001
RCA TFC		35.57 ± 3.48	35.46 ± 3.52	34.22 ± 2.17	37.67 ± 3.28	0.053
Mean TFC		38.01 ± 2.4	37.93 ± 2.18	35.52 ± 1.78	40.67 ± 2.51	<0.001

Data were shown as *n* (%) and mean ± standard deviation. ^a,b^: No statistically significant difference was found between groups with the same letter. HDL-C: High-density lipoprotein cholesterol, LDL-C: Low-density lipoprotein cholesterol, LAD: Left anterior descending artery, CX: Circumflex artery, RCA: right coronary artery. TFC: TIMI frame counts (Frames). * Corrected TIMI frame count is given for LAD. mg: milligram dL: deciliter g: gram.

**Table 2 medicina-59-00763-t002:** Comparisons of the demographic and clinical characteristics of patients who died of cardiovascular vs. non-cardiovascular causes.

		Cause of Death		*p*
		Non-Cardiovascular (*n* = 9)	Cardiovascular (*n* = 12)	
Gender (Male)		6 (66.7)	10 (83.3)	0.611
Hypertension		7 (77.8)	12 (100)	0.171
Diabetes mellitus		6 (66.7)	8 (66.7)	1.000
Family Historyof atherosclerosis		0 (0)	1 (8.3)	1.000
Non-compliance with medical treatment		7 (77.8)	10 (83.3)	1.000
Hyperlipidaemia		1 (11.1)	2 (16.7)	1.000
Smoking Status	Non-smoker	6 (66.7)	5 (41.7)	0.387
	Smoker	3 (33.3)	7 (58.3)	
Intervention	None	9 (100)	10 (83.3)	0.486
	Angiography + stenting	0 (0)	2 (16.7)	
Age (years)		58.67 ± 8.87	60.33 ± 6.83	0.632
Duration of smoking (pack-years)		31 ± 3.61 (*n* = 3)	29.29 ± 4.03 (*n* = 7)	0.545
Total cholesterol (mg/dL)		220.31 ± 38.73	201.6 ± 59.75	0.424
HDL-C (mg/dL)		39.33 ± 7.6	34.83 ± 5.27	0.125
Triglyceride (mg/dL)		181.56 ± 32.66	195.08 ± 24.16	0.288
LDL-C (mg/dL)		144.67 ± 38.74	127.75 ± 55.07	0.442
Hemoglobin (g/dL)		14.12 ± 0.75	14.16 ± 1.77	0.955
Glucose (mg/dL)		99.11 ± 10.02	116.75 ± 39.37	0.207
Survival time (Years)		7.11 ± 2.32	5.42 ± 2.84	0.161
LAD Frame count ***		37 ± 2.92	45.17 ± 3.43	<0.001
CX Frame count		35.33 ± 2.06	39.17 ± 2.72	0.002
RCA Frame count		34.22 ± 2.17	37.67 ± 3.28	0.013
Mean Frame count		35.52 ± 1.78	40.67 ± 2.51	<0.001

Data were shown as *n* (%) and mean ± standard deviation. HDL-C: High-density lipoprotein cholesterol, LDL-C: Low-density lipoprotein cholesterol, LAD: Left anterior descending artery, CX: Circumflex artery, RCA: right coronary artery, TFC: TIMI frame counts (Frames). * Corrected TIMI frame count is given for LAD. mg: milligram dL: deciliter g: gram.

**Table 3 medicina-59-00763-t003:** Multiple logistic regression model for total mortality in patients with CSF.

	*p*	OR	95% CI for OR
Age (years)	0.001	1.220	1.085–1.371
HT	0.022	18.253	1.515–219.851
DM	0.056	5.948	0.953–37.143
Non-compliance with medical treatment	<0.001	75.244	9.671–585.460
HDL-C (mg/dL)	0.019	0.853	0.747–0.975
Triglyceride (mg/dL)	0.575	1.004	0.989–1.02
Constant	0.072		

HDL-C: High-density lipoprotein cholesterol HT: hypertension DM: diabetes mellitus mg: milligram. dL: deciliter.

**Table 4 medicina-59-00763-t004:** Multiple logistic regression model for mortality caused by cardiovascular events in patients with CSF.

	*p*	OR	95% CI for OR
Mean Frame Counts (frames)	0.033	3.318	1.101–10.000
Gender	0.483	7.812	0.025–2451.148
Total cholesterol (mg/dL)	0.287	0.978	0.938–1.019
Constant	0.060	0.000	

mg: milligram dL: deciliter.

## Data Availability

The data presented in this study are available on request from the corresponding author.

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
