# Peer review of "Predictors of Long-Term Mortality in Patients with Stable Angina Pectoris and Coronary Slow Flow"

_medicina, 2023, doi:10.3390/medicina59040763_

Round 1

Reviewer 1 Report

Askoy S et al V investigated the causes of long-term (10-year) mortality in patients with stable angina pectoris (SAP) and coronary slow flow (CSF). A total of 137 patients with CSF were studied. Twenty-one patients (15.3%) died within 10 years of follow-up. Nine (7.2%) and 12 (9.4%) patients died of non-CV and CV causes, respectively. Total mortality in patients with CSF was associated with age, HT, discontinuation of medications, and high-density lipoprotein cholesterol (HDL-C) levels. The mean TFC was associated with CV mortality. The authors concluded that CSF patients exhibited high CV and total mortality after 10 years of follow-up. HT, discontinuation of medications, HDL-C levels, and mean TFC were associated with mortality in CSF patients.

This manuscripts give some new informations about literature data and after, minor revision it merits publication.

1) The authors should discuss why diabtes did not influece CFS (please cited:

DOI: 10.1186/s12933-021-01384-6) 2) the author should stress that CFS is a simplemarkers at coronary angiography with a low-cost (please cite: DOI: 10.1186/1476-7120-6-21) 3) tha authors should give some results in tables.

Author Response

REVIEWER #1 : 1) This manuscripts give some new informations about literature data and after, minor revision it merits publication.The authors should discuss why diabtes did not influece CFS (please cited: DOI: 10.1186/s12933-021-01384-6) 2) the author should stress that CFS is a simplemarkers at coronary angiography with a low-cost (please cite: DOI: 10.1186/1476-7120-6-21) 3) tha authors should give some results in tables.

Response: We agree with the reviewer. We have added the references suggested by the reviewer to the article, and conducted a discussion on these topics in the discussion section.

Reviewer 2 Report

This article is about an interesting aim to determine the causes of long-term (10-year) mortality in patients with stable angina pectoris and coronary slow flow.

However, I have two critical remarks regarding this article:

1. No tables provided for data analysis

2. 36 sources of literature out of 39 older than 5 years ago

Not critical remarks: this is superfluous “P-values < 0.05 were considered significant” in the section methods in abstract.  

All sections of the article are written clearly and concisely. The article can be considered for publication only after consideration of the above comments.

Author Response

REVIEWER #2 : I have two critical remarks regarding this article:

  1. No tables provided for data analysis

Response 1: We agree with the reviewer. When we first uploaded the article to the system, it contained tables and a figure, but we suspect that due to a system error, the tables and figure were not uploaded. Therefore, we apologize for this. We immediately re-added the tables afterwards.

  1. 36 sources of literature out of 39 older than 5 years ago

Response 2:  We replaced most of the references with newly dated ones and added new references.

  1. Not critical remarks: this is superfluous “P-values < 0.05 were considered significant” in the section methods in abstract.  

All sections of the article are written clearly and concisely. The article can be considered for publication only after consideration of the above comments.

Response 3: We removed this sentence from method section in abstract.

Reviewer 3 Report

In this paper, Sukru Aksoy et al found that patients with CSF at coronary angiography, without significant stenosis, exhibited high CV and total mortality at 10-years follow-up. Moreover, they found that HT, discontinuation of medications, HDL-C levels, and mean TFC were the factors associated with mortality.

Even if it is a very interesting and well-written paper, there are some major and minor comments to address.

Major comments

1)      In the introduction section you should clarify if CSF is more common during urgent angiography (STEMI, high risk NSTEMI) or in angiography performed in patients with chronic coronary syndromes.

2)      In the methods, the sentence “A total of 137 patients (93 males) with angiographically normal coronary arteries and CSF were included in this study.” should go in the results section.

3)      You should clarify how did you categorize the causes of death.

4)      You should add the number of protocol and year of the Ethic approval,

5)      Please, clarify why the access site was the femoral artery in all cases.

6)      Were the two independent experienced interventional cardiologists blinded?

7)      To better explain the landmarks, you could put a figure for the three coronary arteries.

8)      Please, include in the results also the percentage of males.

9)      I don’t see any tables attached. Please, provide. Also a figure.

10) In the discussion you stated "In addition, sudden cardiac death due to ventricular arrhythmias has been reported in patients with CSF". Please, provide a percentage to compare to your results.

Minor comments

 1)      It is better to say “patients with CSF” instead of “CSF patients”. Please, correct it through the text.

 2)      Etiology instead of aetiology.

Author Response

REVIEWER #3 : Major comments

About the tables: The authors put 4 tables with summary characteristics of the populations. They should try to combine them into one or a maximum of two tables, putting: general population, alive, dead

 from CV causes, and dead from non-CV causes, and add two p-values (one for comparison between the three groups and another for comparison  between CV and non-CV causes. The rest should only be mentioned in the results without adding further tables.

Response 1: We combined the first four tables (Table1-4) to create two tables (Table 1 and Table2).

1)      In the introduction section you should clarify if CSF is more common during urgent angiography (STEMI, high risk NSTEMI) or in angiography performed in patients with chronic coronary syndromes.

Response 1: We agree with the reviewer. We added the information about CSF is more common during in patients who undergo angiography for acute coronary syndrome, in the introduction section.

2)      In the methods, the sentence “A total of 137 patients (93 males) with angiographically normal coronary arteries and CSF were included in this study.” should go in the results section.

Response 2: This sentence was moved to the methods section of the article.

3)      You should clarify how did you categorize the causes of death.

Response 3: Causes of deaths were extracted from the hospital’s electronic medical records, death certificates and the national health system database. We explained in the methods section how the deaths were categorized.

4)      You should add the number of protocol and year of the Ethic approval,

Response 4: We added the number of protocol and year of the Ethic approval in the material and method section

5)      Please, clarify why the access site was the femoral artery in all cases.

Response 5: We explained why the access site was the femoral artery in all cases in the methods section. Ten years ago, all cases in our hospital were performed through femoral access for economic reasons.

6)      Were the two independent experienced interventional cardiologists blinded?

Response 6: Yes, the two independent experienced interventional cardiologists were blinded. We added this information in the manuscript.

7)      To better explain the landmarks, you could put a figure for the three coronary arteries.

Response 7: We agree with the reviewer, We added two figures to our article, which helped us to better explain these landmark points.

8)      Please, include in the results also the percentage of males.

Response 8: We added this information in the results section

9)      I don’t see any tables attached. Please, provide. Also a figure.

Response 9: When we first uploaded the article to the system, it contained tables and a figure, but we suspect that due to a system error, the tables and figure were not uploaded. Therefore, we apologize for this. We immediately re-added the tables afterwards.

10) In the discussion you stated "In addition, sudden cardiac death due to ventricular arrhythmias has been reported in patients with CSF". Please, provide a percentage to compare to your results.

Response 10:

 We agree with the reviewer, but unfortunately we don’t have any systematic review to compare the percentages. In our study group only one patients (4,8%), died of ventricular arrhythmias.

Minor comments;

 1)      It is better to say “patients with CSF” instead of “CSF patients”. Please, correct it through the text.

Response: We agree with the reviewer, we corrected these terms in the manuscript.

 2)      Etiology instead of aetiology.

Response: we corrected this terms in the manuscript.

Round 2

Reviewer 2 Report

Dear authors, thank you for your work on the article!

Note 1 corrected, everything is in order.

However, remark 2 needs to be improved, since out of 52 sources of literature, half of the sources are still not relevant. Perhaps the authors should reduce the sources of literature by removing some of the irrelevant sources.

Author Response

Response : We agree with the reviewer. We are deeply sorry for the mistakes related to the references. We believe that during the revision of the manuscript, we must have shifted the reference order numbers, which resulted in the references being evaluated as irrelevant. We have removed some of the references from the article and re-organised them as you suggested.

Reviewer 3 Report

The authors addressed all my comments.

Author Response

We greatly appreciate the time, effort, and substantive comments of the reviewer 3.
